# Information is Power:
# Intrinsic Control via Information Capture

**Nicholas Rhinehart**
UC Berkeley

**Jenny Wang**
UC Berkeley

**Glen Berseth**
UC Berkeley

**John D. Co-Reyes**
UC Berkeley

**Danijar Hafner**
University of Toronto
Google Research, Brain Team

**Chelsea Finn**
Stanford University

**Sergey Levine**
UC Berkeley

## Abstract

Humans and animals explore their environment and acquire useful skills even in the absence of clear goals, exhibiting intrinsic motivation. The study of intrinsic motivation in artificial agents is concerned with the following question: what is a good general-purpose objective for an agent? We study this question in dynamic partially-observed environments, and argue that a compact and general learning objective is to minimize the entropy of the agent's state visitation estimated using a latent state-space model. This objective induces an agent to both gather information about its environment, corresponding to reducing uncertainty, and to gain control over its environment, corresponding to reducing the unpredictability of future world states. We instantiate this approach as a deep reinforcement learning agent equipped with a deep variational Bayes filter. We find that our agent learns to discover, represent, and exercise control of dynamic objects in a variety of partially-observed environments sensed with visual observations without extrinsic reward.

## 1 Introduction

Reinforcement learning offers a framework for learning control policies that maximize a given measure of reward – ideally, rewards that incentivize simple high-level goals, such as survival, accumulating a particular resource, or accomplishing some long-term objective. However, extrinsic rewards may be insufficiently informative to encourage an agent to explore and understand its environment, particularly in partially observed settings where the agent has a limited view of its environment. A generalist agent should instead acquire an understanding of its environment before a specific objective or reward is provided. This goal motivates the study of *self-supervised* or *unsupervised* reinforcement learning: algorithms that provide the agent with an intrinsically-grounded drive to acquire understanding and control of its environment in the absence of an extrinsic reward signal. Agents trained with intrinsic reward signals might accomplish tasks specified via simple and sparse rewards more quickly, or may acquire broadly useful skills that could be adapted to specific task objectives. Our aim is to design an embodied agent and a general-purpose intrinsic reward signal that leads to the agent controlling partially-observed environments when equipped only with a high-dimensional sensor (camera) and no prior knowledge.

A large body of prior methods for self-supervised reinforcement learning focus on attaining *coverage*, typically through novelty-seeking or skill-discovery. As argued in prior work [18, 16, 14, 6], a compelling alternative to coverage suited to complex and dynamic environments is to minimize surprise, which incentivizes an agent to control aspects of its environment to achieve homeostasis within it – i.e. constructing and maintaining a niche where it can reliably remain despite external perturbations. We generally expect agents that succeed at minimizing surprise in complex environments to develop

Time

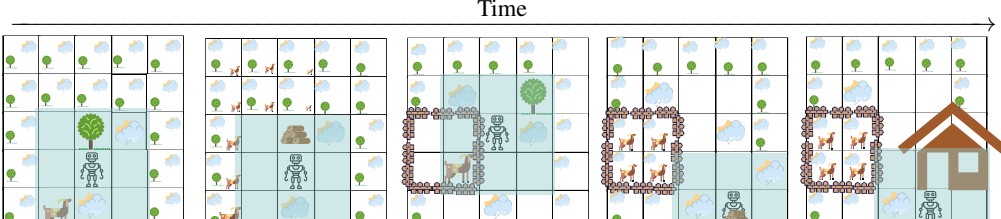

Figure 1: The agent uses a latent state space model to represent beliefs about the world, including dynamic objects like the goat. The blue window represents the agent's field-of-view, which defines the extent of the current observation. Each observation informs the agent's beliefs about the world. In the first panel, the agent sees a tree, a goat, and clouds. In the second panel, the agent chopped down a tree and the goat moved out of view, causing the agent's belief about the goat's position to become uncertain. If the agent reduces the long-horizon average entropy of its beliefs, it will first seek out information (e.g. find the goat), and then modify the environment to reduce changes in belief (e.g. build a fence around the goat). Finally, the agent builds a house to create an area with few changes in weather.

similarly complex behaviors; such acquired complex behaviors may be repurposed for other tasks [6]. However, these frameworks do not explicitly address the difficulty of controlling partially-observed environments: if an otherwise complex and chaotic environment contains a "dark room" (small reliable niche), an agent could minimize surprise simply by hiding in this room and refusing to make meaningful observations, thereby failing to explore and control the wider surrounding environment.

Consider Fig. 1, which depicts a partially-observed outdoor environment with trees, a goat, weather, and an agent. We will discuss three different intrinsic incentives an agent might adopt in this environment. If the agent's incentive is to (*i*) minimize the entropy of its next observation, it will seek the regions with minimal unpredictable variations in flora, fauna, and weather. This is unsatisfying because it merely requires avoidance, rather than interaction. Let us assume the agent will maintain a model of its *belief* about a learned *latent state* – the agent cannot observe the *true* state, instead it learns a state representation. Further, let us assume the agent maintains a separate model of the visitation of its latent state – we will refer to this distribution as its *latent visitation*. If the agent's incentive is to (*ii*) minimize the entropy of belief (either at every step or at some final step), the agent will gather information and take actions to make the environment predictable: find and observe the changes in flora, fauna, and weather that are predictable and avoid those that aren't. However, once it has taken actions to make the world predictable, this agent is agnostic to future change – it will not resist predictable changes in the environment. Finally, if the agent's incentive is to (*iii*) minimize the entropy of its latent visitation, this will result in categorically different behavior: the agent will seek both to make the world predictable by gathering information about it and *prevent it from changing*. While both the belief and latent visitation entropy minimization objectives are worthwhile intrinsic motivation objectives to study, we speculate that an agent that is adept at preventing its environment from changing will generally learn more complex behaviors.

We present a concise and effective objective for self-supervised reinforcement learning in dynamic partially-observed environments: minimize the entropy of the agent's latent visitation under a latent state-space model learned from exploration. Our method, which we call Intrinsic Control by Information Capture (IC2), causes an agent to learn to seek out and control factors of variation outside of its immediate observations. We instantiate this framework by learning a state-space model as a deep variational Bayes filter along with a policy that employs the model's beliefs. Our experiments show that our method learns to discover, represent, and control dynamic entities in partially-observed visual environments with *no extrinsic reward signal*, including in several 3D environments.

**Maxwell's Demon and belief entropy**   The main concept behind our approach to self-supervised reinforcement learning is that incentivizing an agent to minimize the entropy of its *beliefs* about the world is sufficient to lead it to *both* gather information about the world *and* learn to control aspects of its world. Our approach is partly inspired by a well-known connection between information theory and thermodynamics, which can be illustrated informally by a version of the Maxwell's Demon thought experiment [46, 41]. Imagine a container separated into compartments, as shown in Fig. 2. Both compartments contain gas molecules that bounce off of the walls and each other in a somewhat unpredictable fashion, though short-term motion of these molecules

(between collisions) is predictable. The compartments are separated by a massless door, and the agent (the eponymous "demon") can open or close the door at will to sort the particles.[1]

By sorting the particles onto one side, the demon appears to reduce the disorder of the system, as measured by the thermodynamic entropy, $S$, which increases the energy, $F$ available to do work, as per Helmholtz's free energy relationship, $F = U - TS$. The ability to do work affords the agent *control over the environment*. This apparent violation of the second law of thermodynamics is resolved by accounting for the agent's information processing needed to make decisions [5]. By concentrating the particles into a smaller region, the number of states each particle visits is reduced. This illustrates an example environment in which reducing the entropy of the visitation distribution results in an agent gaining the

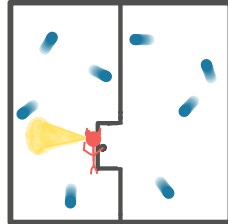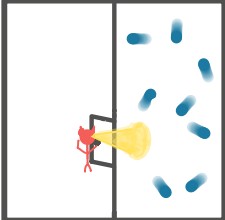

Figure 2: A "demon" gathering information to sort particles, reducing the entropy of the particle configuration.

ability to do work. In the same way that Maxwell's demon accumulates free energy via information-gathering and manipulating its environment, we expect self-supervised agents guided by belief visitation entropy minimization to accumulate the equivalent of potential energy in their corresponding sequential decision processes, which leads them to gain control over their environments.

**Preliminaries** Our goal in this work will be to design self-supervised reinforcement learning methods in partially observed settings to acquire complex behaviors that both gather information and gain control over their environment. To this end, we will formulate the learning problem in the context of a discrete-time partially-observed controlled Markov process, also known as a controlled hidden Markov process (CHMP), which corresponds to a POMDP without a reward function. The CHMP is defined by a state space $\mathcal{S}$ with states $\mathbf{s} \in \mathcal{S}$, action space $\mathcal{A}$ with actions $\mathbf{a} \in \mathcal{A}$, transition dynamics $P(\mathbf{s}_{t+1}|\mathbf{s}_t, \mathbf{a}_t)$, observation space $\Omega$ with observations $\mathbf{o} \in \Omega$, and emission distribution $O(\mathbf{o}_t|\mathbf{s}_t)$. The agent is a policy $\pi(\mathbf{a}_t|\mathbf{o}_{\leq t})$. Note that it does *not* observe any states $\mathbf{s}$.

We denote the undiscounted finite-horizon state visitation as $d^\pi(\mathbf{s}) \doteq 1/T \sum_{t=0}^{T-1} \Pr_\pi(\mathbf{s}_t = \mathbf{s})$, where $\Pr_\pi(\mathbf{s}_t = \mathbf{s})$ is the probability that $\mathbf{s}_t = \mathbf{s}$ after executing $\pi$ for $t$ steps. Using $d^\pi(\mathbf{s})$, we can quantify the average *disorder* of the environment with the Shannon entropy, $H(d^\pi(\mathbf{s}))$. Prior work proposes surprise minimization ($\min_\pi - \log \hat{p}(\mathbf{s})$) as an intrinsic control objective [15, 70, 6]; in Berseth et al. [6] (SMiRL), the agent models the state visitation distribution, $d^\pi(\mathbf{s})$ with $\hat{p}(\mathbf{s})$, which it computes by assuming access to $\mathbf{s}$. In environments in which there are natural sources of variation outside of the agent, this incentivizes the SMiRL agent, fully aware of these variations observed through $\mathbf{s}$, to take action to control them. In a partially-observed setting, it is a matter of interpretation of what the most natural extension of SMiRL is – whether (1) the observation should be treated as the state, or (2) an LSSM's belief should be treated as an estimate of the latent state. Thus, our experiments include both interpretations, the latter constituting one of our methods. In the former, SMIRL's model becomes $\hat{p}(\mathbf{o})$, which generally will enable the agent to ignore variations that it can prevent from appearing in its observations. We observe this phenomenon in our experiments.

## 2 Control and Information Gathering via Belief Entropy Minimization

The main question that we tackle in the design of our algorithm is: how can we formulate a general and concise objective function that can enable an RL agent to gain control over its partially-observed environment, in the absence of any user-provided task reward? Consider the following partially-observed "TwoRoom" environment, depicted in Fig. 3. The environment has two rooms: an empty ("dark") room on the left, and a "busy" room on the right, the latter containing two moving particles that move around until the agent "tags" them, which stops their motion. Intuitively, an agent that aims to gather information and gain control of its environment should search for the moving particles to find out where they are. However, it is difficult to observe both particles at the same time. A more effective strategy is to "tag" the particles – then, their position remains fixed, and the agent will know where they are at all times, even when they are not observed. This task, remininscent of the

---

[1]In Maxwell's original example, the demon sorts the particles into the two chambers based on velocity. Our example is actually more closely related to Szilard's engine, which is based on a similar principle [68, 45].

Time

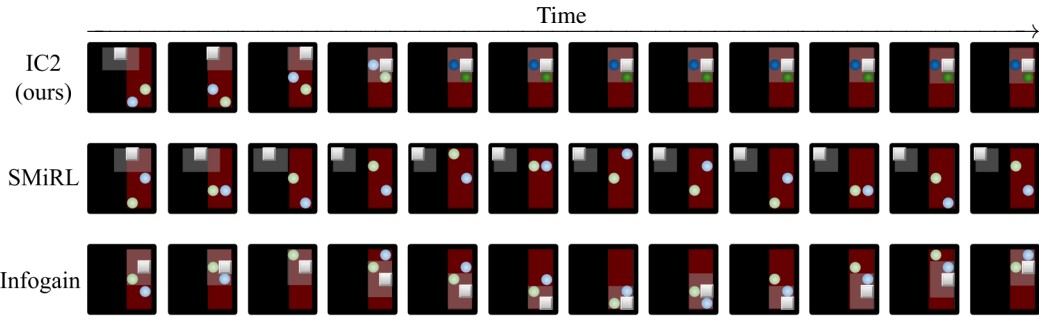

IC2
(ours)

SMiRL

Infogain

Figure 3: Comparison of intrinsic control objectives on the TwoRoom environment. The agent, in white, can view a limited area around it, in grey, and can stop particles within its view and darken their color. The vertical wall, in brown, separates particles (blue and green) in the "busy room" (on right) from the "dark room" (on left). *Top*: Our approach seeks out the particles and stops them. *Middle:* The observational surprise minimization method in Berseth et al. [6] leads the agent to frequently hide in the dark room, leaving the particles unstopped. *Bottom:* Latent-state infogain leads the agent to find and observe the particles, but not stop them.

previously-discussed Maxwell's Demon thought experiment, can be seen as a simple analogy for more complex settings that can occur in natural environments.

**Representing variability with latent state-space models.** In order for the agent to represent the dynamic components of the environment observed from images, our method involves learning a latent state-space model (LSSM) [72, 38, 29, 44, 22, 47, 73, 71, 40, 10, 23, 48, 58]. We intermittently refer to these dynamic components as "factors of variation" to distinguish the model's representation of variability in the environment (latent state) from the true variability (true state). At timestep $t$, the LSSM represents the agent's current *belief* as $q_\phi(\mathbf{z}_t|\mathbf{o}_{\leq t}, \mathbf{a}_{\leq t-1})$, where $\mathbf{z}_t$ is the model's latent state. We defer the description of the LSSM learning and architecture to Section 3, and now motivate how we will use the LSSM for constructing an intrinsic control objective.

**Belief entropy and latent visitation entropy.** Consider a policy that takes actions to minimize the entropy of the belief $H(q_\phi(\mathbf{z}_t|\mathbf{o}_{\leq t}, \mathbf{a}_{\leq t-1}))$. This corresponds to the agent performing *active state estimation* [12, 74, 37], and is equivalent to taking actions to maximize expected latent-state information gain $I(\mathbf{o}_t, \mathbf{z}_t|\mathbf{o}_{<t}, \mathbf{a}_t)$ [2]. However, active state estimation is satisfied by a policy that simply collects informative observations, as it does not further incentivize actions to "stabilize" the environment by preventing the latent state from changing. Analogous to the standard definition of undiscounted state visitation, consider the undiscounted latent visitation: $d^\pi(\mathbf{z}) \doteq 1/T \sum_{t=0}^{T-1} \Pr_\pi(\mathbf{z}_t = \mathbf{z})$, where $\Pr_\pi(\mathbf{z}_t = \mathbf{z}) = \mathbb{E}_\pi q_\phi(\mathbf{z}_t|\mathbf{o}_{\leq t}, \mathbf{a}_{\leq t-1})$ (the expected belief after executing $\pi$ for $t$ timesteps). Our goal is to minimize $H(d^\pi(\mathbf{z}))$, as this corresponds to *stabilizing the agent's beliefs*, which incentivizes both reducing uncertainty in each $q_\phi(\mathbf{z}_t|\mathbf{o}_{\leq t}, \mathbf{a}_{\leq t-1})$, as well as constructing a niche such that each $q_\phi(\mathbf{z}_t|\mathbf{o}_{\leq t}, \mathbf{a}_{\leq t-1})$ concentrates probability on the same latent states. While in principle we could approximate this entropy and minimize it directly, later we discuss our more practical approach.

**Discovering factors of variation.** In order for belief entropy minimization to incentivize the agent to control entities in the environment, the LSSM's belief must represent the underlying state variables in some way and model their uncertain evolution until either observed or controlled. For example, the demon in the thought experiment in Section 1 would have no incentive to gather the particles if it did not know that they existed. If the agent is unable to encounter or represent dynamic objects in its LSSM, it could choose not to move. However, if the LSSM properly captures the existence and motion of dynamic objects (even out of view), then the agent has an incentive to exert control over them, rather than avoid them by going to parts of the environment where it does not encounter them. While sufficient random exploration may result in a good-enough LSSM, making this approach generally practical requires a suitable exploration strategy to collect the experience necessary to train an LSSM that represents all of the underlying factors of variation.

To this end, we learn a separate exploratory policy to maximize *expected model information gain*, similar to Schmidhuber [60], Houthooft et al. [26], Gheshlaghi Azar et al. [20], Sekar et al. [62]. The exploration policy is employed as a way to encounter dynamic objects and collect data on which the model performs poorly. The exploration policy does not need to learn accurate estimates of the latent state and its uncertainty – it is merely incentivized to visit where the current model is inaccurate.

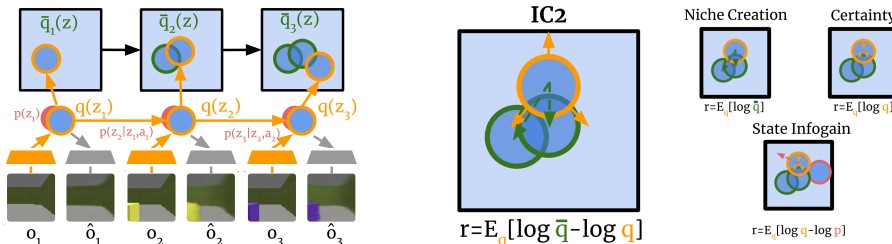

(a) Latent state-space and visitation models.     (b) Intrinsic rewards.

Figure 4: Figure of latent-state space model and rewards. *Left:* The model observes images, $\mathbf{o}_t$ to inform beliefs about latent states, $q_\phi(\mathbf{z}_t|\mathbf{o}_{\leq t}, \mathbf{a}_{\leq t-1})$, and observes actions to make one-step predictions $p(\mathbf{z}_{t+1}|\mathbf{z}_t, \mathbf{a}_t)$. Each belief is used to update the latent visitation, $\bar{q}_{t'}(\mathbf{z})$. *Right:* The beliefs and latent visitations can be combined into various reward functions. We denote our main method's reward function "IC2". The solid arrows denote directions of belief expansion and contraction incentivized by rewards; the dotted arrows denote directions of belief translation incentivized by rewards.

Expected information gain about model parameters $\boldsymbol{\theta}$ is relative to a set of prior experience $\mathcal{D}$ and a partial trajectory $\mathbf{h}_{t-1}$, given as $I(\mathbf{o}_t; \boldsymbol{\theta}|\mathbf{h}_{t-1}, \mathcal{D}) = \mathbb{E}_{\mathbf{o}_t} \mathrm{KL}(p(\boldsymbol{\theta}|\mathbf{o}_t, \mathbf{h}_{t-1}, \mathcal{D}) \,||\, p(\boldsymbol{\theta}|\mathbf{h}_{t-1}, \mathcal{D}))$. Note that model information gain is *distinct* from the information an agent may gather to reduce its belief entropy $H(q_\phi(\mathbf{z}_t|\mathbf{o}_{\leq t}, \mathbf{a}_{\leq t-1}))$ within the current episode. Computing the full model parameter prior $p(\boldsymbol{\theta}|\mathbf{h}_{t-1}, \mathcal{D})$ and posterior $p(\boldsymbol{\theta}|\mathbf{o}_t, \mathbf{h}_{t-1}, \mathcal{D})$ is generally computationally expensive, and also requires evaluating an expectation over observations – instead, we approximate this expected information gain following a method similar to Sekar et al. [62]: we use an ensemble of latent dynamics models, $\mathcal{E} = \{p_{\theta_i}(\mathbf{z}_t|\mathbf{z}_{t-1}, \mathbf{a}_{t-1})\}_{i=1}^{K}$ to compute the variance of latent states estimated by $q_\phi(\mathbf{z}_t|\mathbf{o}_{\leq t}, \mathbf{a}_{\leq t-1})$. We build the ensemble throughout training using the method of Izmailov et al. [27]. Thus, the exploration reward is given as: $r_e = \mathrm{Var}_{\{\theta_i\}}[\log p_\theta(\mathbf{z}_t|\mathbf{z}_{t-1}, \mathbf{a}_{t-1})|\mathbf{z}_t \sim q_\phi]$.

# 3   The IC2 Algorithm

Now we describe how we implement our intrinsic control algorithm. The main components are the latent-state space model, the latent visitation model, and the exploration and control policies.

**Latent state-space model.**    In our CHMP setting, the agent only has access to partial observations $\mathbf{o}$ of the true state $\mathbf{s}$. In order to estimate a representation of states and beliefs, we employ a sequence-based Variational Autoencoder to learn latent variable belief, dynamics, and emission models. We formulate the variational posterior to be $q_\phi(\mathbf{z}_{1:T}|\mathbf{o}_{1:T}, \mathbf{a}_{1:T}) = \prod_{t=1}^{T} q_\phi(\mathbf{z}_t|\mathbf{o}_{\leq t}, \mathbf{a}_{\leq t-1})$ and the generative model to be $p(\mathbf{z}_{1:T}, \mathbf{o}_{1:T}|\mathbf{a}_{1:T}) = \prod_{t=1}^{T} p_\theta(\mathbf{o}_t|\mathbf{z}_t)p_\theta(\mathbf{z}_t|\mathbf{z}_{t-1}, \mathbf{a}_{t-1})$. Denoting $\mathbf{h}_t \doteq (\mathbf{o}_{\leq t}, \mathbf{a}_{\leq t-1})$, the log-evidence of the model and its lower-bound are:

$$\log p(\mathbf{o}_{1:T}|\mathbf{a}_{1:T-1}) = \log \mathbb{E}_{\mathbf{z}_{1:T} \sim p(\mathbf{z}_{1:T}|\mathbf{a}_{1:T})} \prod_{t=1}^{T} \log p(\mathbf{o}_t|\mathbf{z}_t) \geq \mathcal{L}(\phi, \theta; \mathbf{o}_{1:T}, \mathbf{a}_{1:T-1}^i)$$

$$= \sum_{t=1}^{T} \mathbb{E}_{q_\phi(\mathbf{z}_t|\mathbf{h}_t)} \left[ p_\theta(\mathbf{o}_t|\mathbf{z}_t) \right] - \mathbb{E}_{q_\phi(\mathbf{z}_{t-1}|\mathbf{h}_{t-1})} \left[ \mathrm{KL}(q_\phi(\mathbf{z}_t|\mathbf{h}_t) \,||\, p_\theta(\mathbf{z}_t|\mathbf{z}_{t-1}, \mathbf{a}_{t-1})) \right]. \quad (1)$$

Given a dataset $\mathcal{D} = \{(\mathbf{o}_{1:T}, \mathbf{a}_{1:T})_i\}_{i=1}^{N}$, Eq. 1 is used to train the model via $\max_{\phi,\theta} 1/|\mathcal{D}| \sum_i \mathcal{L}(\phi, \theta; \mathbf{o}_{1:T}^i, \mathbf{a}_{1:T-1}^i)$. The focus of our work is not to further develop the performance of LSSMs, but improvements in the particular LSSM employed would translate to improvements in our method. In practice, we implemented an LSSM in PyTorch [55] similar to the categorical LSSM architecture described in [24]. In this case, both the belief prior and belief posterior are distributions formed from products of $K_1$ categorical distributions over $K_2$ categories: $g(\mathbf{z}; \mathbf{v}) = \prod_{\kappa_1=1}^{K_1} \prod_{\kappa_2=1}^{K_2} \mathbf{v}_{\kappa_1, \kappa_2}^{\mathbf{z}_{\kappa_1, \kappa_2}}$, where the vector of $K_1 \cdot K_2$ parameters, $\mathbf{v}$, is predicted by neural networks: $\mathbf{v}_{\text{posterior}} = f_\phi(\mathbf{o}_{\leq t}, \mathbf{a}_t)$ for the posterior, and $\mathbf{v}_{\text{prior}} = f_\theta(\mathbf{o}_{<t}, \mathbf{a}_t)$ for the prior. We found it effective to construct $f_\phi$ and $f_\theta$ following the distribution-sharing decomposition described in [29, 30, 10], in which the posterior is produced by transforming the parameters of the prior with a measurement update. We implemented the prior as an RNN, and the posterior as an MLP, and defer further details to the appendix.

**Latent visitation model.**    Our agent cannot, in general, evaluate $d^\pi(\mathbf{z})$ or $H(d^\pi(\mathbf{z}))$; at best, it can approximate them. We do so by maintaining a within-episode estimate of $d^\pi(\mathbf{z})$ by constructing a

---

**Algorithm 1** Intrinsic Control via Information Capture (IC2)

---

1: **procedure** IC2(Env; $K, M, N, L$)
2:   Initialize $\pi_c, \pi_e, q_\phi, p_{\{\theta_i\}_{i=1}^K}, \mathcal{D} \leftarrow \emptyset$.
3:   **for** episode $= 0, \dots, M$ **do**
4:     $\mathcal{D}_e \leftarrow \text{Collect}(N, \text{Env}, \pi_e); \mathcal{D}_c \leftarrow \text{Collect}(N, \text{Env}, \pi_c); \mathcal{D} \leftarrow \mathcal{D} \cup \mathcal{D}_e \cup \mathcal{D}_c$
5:     Update $q_\phi, p_\theta$ using SGD on Eq. 1 with data $\mathcal{D}$ for $L$ rounds.
6:     Update $\pi_e$ using SGD on PPO's objective with rewards $r_e$ with data $\mathcal{D}_e$
7:     Update $\pi_c$ using SGD on PPO's objective with the IC2 reward (Eq. (2)) with data $\mathcal{D}_c$

---

mixture across the belief history samples $\bar{q}_{t'}(\mathbf{z}) = {}^1\!/t' \sum_{t=0}^{t'} q_\phi(\mathbf{z}_t|\mathbf{o}_{\leq t}, \mathbf{a}_{\leq t-1})$. This corresponds to a mixture (across time) of single-sample estimates of each $\mathbb{E}_\pi q_\phi(\mathbf{z}_t|\mathbf{o}_{\leq t}, \mathbf{a}_{\leq t-1})$. Given $\bar{q}_{t'}(\mathbf{z})$, we have an estimate of the visitation of the policy, and we can use this as part or all of the reward signal of the agent. To implement $\bar{q}_{t'}(\mathbf{z})$, we found that simply recording each belief sufficed. While we adopt a within-episode estimate of $d^\pi(\mathbf{z})$, our approach could be extended to multi-episode estimation, at the expense of extra computational burden.

### 3.1 The IC2 reward

We now describe our main reward function, "IC2". In what follows, we denote $q_\phi(\mathbf{z}_t|\mathbf{o}_{\leq t}, \mathbf{a}_{\leq t-1}) = q_t(\mathbf{z}_t)$ for brevity. The IC2 reward is the KL from the belief to the belief visitation (Eq. (2)). It is visualized in Fig. 4. This incentivizes the agent to reduce belief visitation entropy by bringing the current belief towards the current latent visitation distribution, and to expand the current belief to encourage exploration, and thus potentially find a broader niche.

$$r_t^{ne} \doteq -\text{KL}(q_t(\mathbf{z}_t)||\bar{q}_{t'}(\mathbf{z})) = \mathbb{E}_{q_t(\mathbf{z}_t)}[\log \bar{q}_{t'}(\mathbf{z}) - \log q_t(\mathbf{z}_t)]. \tag{2}$$

The LSSM and visitation models used in our method allow for easy constructions of other intrinsic reward functions. We construct and experiment with several other intrinsic rewards in order to provide more context for the effectiveness of our main method.

**Niche Creation.** The expected negative surprise of the current belief under the latent visitation distribution measures how unsurprising the current belief is relative to the agent's experience in the episode so far. One interpretation of this reward is that it is SMiRL [6] applied to samples of latent beliefs estimated from a LSSM in a CHMP. Minimizing this reduces the number of visited belief states and increases the agent's certainty. The *niche creation reward* is:

$$r_t^{nc} \doteq -\text{H}(q_t(\mathbf{z}_t), \bar{q}_{t'}(\mathbf{z})) = \mathbb{E}_{q_t(\mathbf{z}_t)}[\log \bar{q}_{t'}(\mathbf{z})]. \tag{3}$$

**Certainty.** The entropy of the current belief measures the agent's certainty of the latent state, and by extension, about the aspects of the environment captured by the latent. Minimizing the belief entropy increases the agent's certainty. It is agnostic to predictable changes, and penalizes unpredictable changes. We define the *certainty reward* as the negative belief entropy:

$$r_t^c \doteq -\text{H}(q_t(\mathbf{z}_t)) = \mathbb{E}_{q_t(\mathbf{z}_t)}[\log q_t(\mathbf{z}_t)] \tag{4}$$

**State Infogain.** The latent state information gain (not the model information gain in Section 2) measures how much more certain the belief is compared to its temporal prior. Gaining information does not always coincide with being certain, because an infogain agent may cause chaotic events in the environment with outcomes that it only understands partially. As a result, it has gained more information than standing still but has also become less certain. We define the *infogain reward* as:

$$r_t^i \doteq \mathbb{E}_{q_{t-1}(\mathbf{z}_{t-1})}\text{KL}(q_t(\mathbf{z}_t)||p(\mathbf{z}_t|\mathbf{z}_{t-1}, \mathbf{a}_{t-1})) = \mathbb{E}_{q_t(\mathbf{z}_t)q_{t-1}(\mathbf{z}_{t-1})}[\log {}^{q_t(\mathbf{z}_t)}\!/p(\mathbf{z}_t|\mathbf{z}_{t-1}, \mathbf{a}_{t-1})]. \tag{5}$$

### 3.2 Algorithm summary

Conceptual pseudocode for for Intrinsic Control via Information Capture (IC2) is presented in Alg. 1. The algorithm begins by initializing the LSSM, $q_\phi$ and $p_\theta$, as well as two separate policies: one trained to collect difficult-to-predict data with the exploration objective with rewards defined by $r_e$, $\pi_e$, and one trained to maximize a intrinsic control objectives defined by one of Eqs. (2) to (5).

We represent each policy $\pi(\mathbf{a}_t|\mathbf{v}_{\text{posterior},t})$ as a two-layer fully-connected MLP with 128 units. Recall that $\mathbf{v}_{\text{posterior}}$ is the vector of posterior parameters, which enables the policy to use the memory

represented by the LSSM. We do not back-propagate the policies' losses to the LSSM for simplicity of implementation, although prior work does so in the case of a single policy [40]. Our method is agnostic to the subroutine used to improve the policies. In our implementation, we employ PPO [61].

## 4 Experiments

Our experiments are designed to answer the following questions: **Q1: Intrinsic control capability**: Does our latent visitation-based self-supervised reward signal cause the agent to stabilize partially-observed visual environments with dynamic entities more effectively than prior self-supervised stabilization objectives? **Q2: Properties of the IC2 reward and alternative intrinsic rewards**: What types of emergent behaviors does each belief-based objective described in Section 3.1 evoke?

In order to answer these questions, we identified environments with the following properties **(i)**: partial observability, **(ii)**: dynamic entities that the agent can affect, and **(iii)**: high-dimensional observations. Because many standard RL benchmarks do not contain the significant partial-observability that is prevalent in the real world, it is challenging to answer Q1 or Q2 with them. Instead, we create several environments, and employ several existing environments we identified to have these properties. In what follows, we give an overview of the experimental settings and conclusions. We defer comprehensive details to the appendix.

**TwoRoom Environment.** As previously described, this environment has two rooms: an empty ("dark") room on the left, and a "busy" room on the right, the latter containing moving particles (colored dots) that move unless the agent "tags" them, which permanently stops their motion, as shown in Fig. 5. The agent (white) observes a small area around it (grey), which it receives as an image. In this environment, control corresponds to finding and stopping the particles. The action space is $\mathcal{A} = \{$left, right, up, down, tag, no-op$\}$, and the observation space is normalized RGB-images: $\Omega = [0, 1]^{3 \times 30 \times 30}$. An agent that has significant control over this environment should tag particles to reduce the uncertainty over future states. To evaluate policies, we use the average

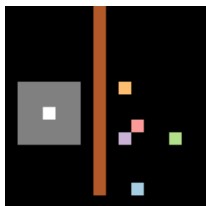

Figure 5: TwoRoom Large Environment.

fraction of total particles locked, the average fraction of particles visible, and the discrete true-state visitation entropy of the positions of the particles, $H(d^\pi(s_d))$. We employed two versions of this environment, with details provided in the appendix. In the large environment, the agent observes a 5x5 area inside a 15x15 area, and the busy room contains 5 particles.

**VizDoom DefendTheCenter environment** The VizDoom DefendTheCenter environment shown in Fig. 6 is a circular arena in which a stationary agent, equipped with a partial field-of-view of the arena and a weapon, can rotate and shoot encroaching monsters [31]. The action space is $\mathcal{A} = \{$turn left, turn right, shoot$\}$, and the observation space is normalized RGB-images: $\Omega = [0, 1]^{3 \times 64 \times 64}$. In this environment, control corresponds to reducing the number of monsters by finding and shooting them. We use the average original environment return, (which *no policy* has access to during training), the average number of killed monsters at the end of an episode, and the average number of visible monsters to measure the agent's control.

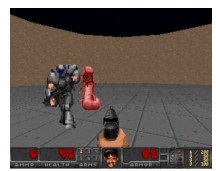

Figure 6: Vizdoom Defend The Center.

**OneRoomCapture3D environment.** The MiniWorld framework is a customizable 3D environment simulator in which an agent perceives the world through a perspective camera [9]. We used this framework to build the environment in Fig. 7 which an agent and a bouncing box both inhabit a large room; the agent can lock the box to stop it from moving if it is nearby, as well as constrain the motion of the box by standing nearby it. In this environment, control corresponds to finding the box and either trapping it near a wall, or tagging it. The action space is $\mathcal{A} = \{$turn left $20°$, turn right $20°$, move forward, move backward, tag$\}$, and observation space is normalized RGB-images: $\Omega = [0, 1]^{3 \times 64 \times 64}$. We use the

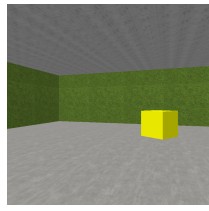

Figure 7: One Room Capture 3D.

average fraction of time the box is captured, the average time the box is visible, and differential entropy (Gaussian-estimated) of the true-state visitation of the box's position $H(d^\pi(s_d))$ to measure the agent's ability to control the environment.

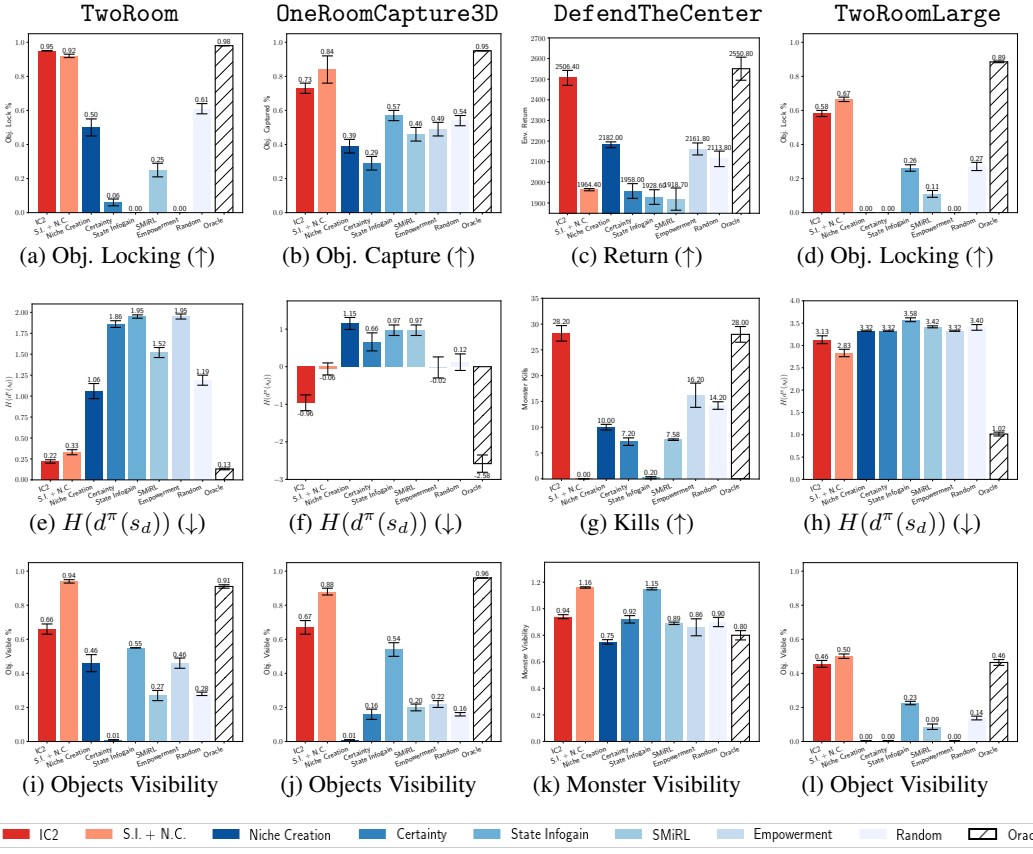

Figure 8: Policy evaluation. Means and their standard errors are reported. The IC2 and Niche Creation+Infogain objectives lead the agent to stabilize the dynamic objects more effectively than other methods.

## 4.1 Comparisons

In order to answer **Q1**, we compare to SMiRL [6] and a recent empowerment method [77] that estimates the current empowerment of the agent from an image. We use the empowerment estimate as a reward signal for training a policy. In order to answer **Q2**, we ensured each environment was instrumented with the aforementioned metrics of visibility and control, and deployed Algorithm 1 separately with each of Eqs. (2) to (5), as well a with simple sum of Eq. (3) and Eq. (5). We also compare to a "random policy" that chooses actions by sampling from a uniform distribution over the action space, as well as an "oracle policy" that has access to privileged information about the environment state in each environment. We perform evaluation at $5e6$ environment steps with 50 policy rollouts per random seed, with 3 random seeds for each method (150 rollouts total). *Exploration:* Note that while our method employs an exploration policy, it is used only to improve the LSSM – the data collected by the exploration policy is not used by the on-policy learning of the control policy (although it could be used, in principle, by an off-policy learning procedure). No method, including ours, is provided with an explicit separate mechanism to augment data collection for policy optimization. All control policies employed in our methods and the baselines follow the same exploration procedure that occurs as part of PPO optimization.

**Oracle policies.** The oracle policies are designed to maximize the control performance in each environment as much as practically possible to put the control performance of our intrinsic objectives in context. Their purpose is to approximate the highest possible performance in each environment, for which access to privileged information can be convenient. In the TwoRoom and OneRoomCapture3D environments, the oracle policies are scripted, using access to the environment state, to seek out the nearest unfrozen object and freeze it, thereby achieving high performance on the lock, capture, and entropy control metrics. In the DefendTheCenter environment, the oracle policy is a neural network

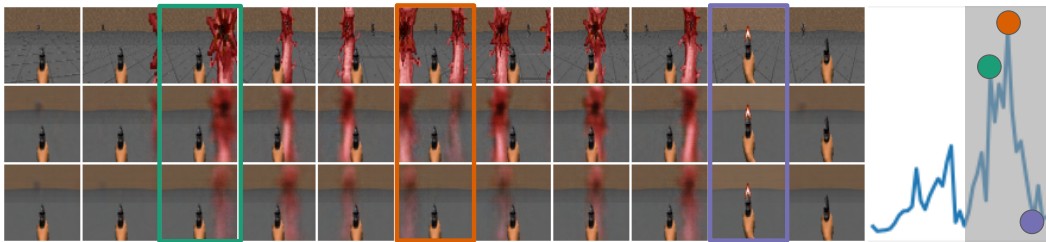

Figure 9: Visualization of a sequence in the VizDoom DefendTheLine environment. *Row 1:* The image provided to the agent. *Row 2:* The agent's reconstruction of a sample from $q$. *Row 3:* The agent's one-step image forecast. *Right:* The state infogain signal, $\mathbb{E}_q[\log q - \log p]$. Each colored rectangle identifies a keyframe that corresponds to a colored circle on the infogain plot. The infogain signal measures how much more certain the belief is compared to its temporal prior; when stochastic events happen (monster appears nearby), the signal is high; when the next image is predictable (monster disappears when shot), the signal is low.

trained using PPO to shoot as many monsters as possible by accessing the true extrinsic task reward of the environment.

**Results summary.** Our primary results are presented in Fig. 8. We observe the IC2 and Niche Creation+Infogain (NCI) rewards to yield policies that exhibit a high degree of control over each environment – finding and stopping the moving objects, and finding and shooting the monsters, in the *complete absence of any extrinsic reward signal*. The results show that employing the NC objective accompanied by a within-episode bonus term – resulting in either the IC2 objective or the NCI objective – achieves the best control performance across the environments tested. The Infogain-based agent generally seeks out and observes, but does not interfere with, the dynamic objects; the high Visibility metric and low control metrics (Lock, Capture, Kill) in each environment illustrates this. Qualitative results of this phenomenon are illustrated in Fig. 9, in which the infogain signal is high when stochastically-moving monsters are visible, and low when they are not. We observe the method of Berseth et al. [6] tends to learn policies that hide from the dynamic objects in the environment, as indicated by the low control and visibility values of the final policy. Furthermore, we observe the method of Zhao et al. [77] not to exhibit controlling behavior in these partially-observed environments, instead it views the dynamic objects in the TwoRoom and OneRoomCapture3D Environments somewhat more frequently than the random policy. We present videos on our project page: https://sites.google.com/view/ic2/home.

**Detailed DefendTheCenter results.** We observe the infogain-based agent to rotate the camera until a close monster is visible and then oscillate the camera centered on the monster. The infogain leads to the agent maintaining a large visual stimulus that is difficult for the model to predict due to the stochastic motion of the monster, an interpretation supported by Fig. 8k, where Infogain and NCI agents have the highest number of visible monsters in their field of view. We also observe that the NCI policy exhibits similar behavior to Infogain, which we speculate may be due to the infogain component of the objective dominating the NC component. We observe the NC agent to minimize visibility of the monsters in the following way: the agent tends to shoot the immediately visible monsters upon first spawning and then toggle between turning left and right. Since there is no no-op action, this allows the agent to see as few monsters as possible, since its view is centered on a region of the map it just cleared. We observe the IC2 agent to typically rotate in one direction and reliably shoot monsters. Our interpretation of this is that the IC2 reward is causing the agent to seek out and reduce the number of surprising monsters more often than the NC agent due to the belief entropy bonus term that IC2 contains, which encourages more exploration compared to NC. The Certainty agent has similar behavior to the NC agent but we observe that it is not as successful and has fewer monster kills and more visible monsters than NC.

**Detailed OneRoomCapture3D results.** Figs. 8b and 8f show that the IC2 scores lower (better) than NCI in the box entropy metric yet slightly worse in the Captured metric due to (1) differences in how each agent captures the box and (2) higher variance in the NCI policy optimization. (1) The IC2 agent tends to moves towards the box and execute the "freeze" action to stop the box from moving, whereas the NCI agent tends to move toward the box and trap it next to the wall without executing the freeze action, which causes the box to bounce back and forth between the wall and the agent every timestep, which is measured as Captured. We hypothesize NCI does this because allowing the

box to retain some movement gives the agent more Infogain reward. (2) Higher variance in the policy optimization of the NCI agent resulted in some runs that performed worse, which impacted both the entropy and capture metrics, but had a stronger effect on the box capturing metric.

## 5   Related Work

Much of the previous work on learning without extrinsic rewards has been based either on (i) exploration [8, 53, 52], or (ii) some notion of intrinsic control, such as empowerment [34, 49, 30]. Exploration approaches include those that maximize model prediction error or improvement [59, 43, 65, 56], maximize model uncertainty [26, 66, 64, 57, 20], maximize state visitation [4, 19, 69, 25], maximize surprise [59, 1, 67], and employ other novelty-based exploration bonuses [42, 7, 32, 33]. Our method can be combined with prior exploration techniques to aid in optimizing our proposed objective, and in that sense our work is largely orthogonal to prior exploration methods.

Prior works on intrinsic control include empowerment maximization [34, 35, 50], observational surprise minimization [15, 17, 70, 6, 54], and skill discovery [3, 36, 21, 11, 63, 75]. Observational surprise minimization seeks policies that make *observations* predictable and controllable, and is closely connected to entropy minimization, as entropy is defined to be the expected surprise. In Friston et al. [18], the notion of Free Energy Minimization corresponds to minimizing *observational* entropy, and states that the entropy of hidden states in the environment is bounded by the entropy of sensory observations. However, the proof assumes a diffeomorphism to hold between states and observations, which is explicitly violated in CHMPs and any real-world setting, as agents cannot perceive the state of anything outside their egocentric sensory observations. Similarly, empowerment [35, 28, 30, 77] is a measure of the degree of control an agent *could have* over future *observations*, whereas state visitation entropy is a measure of the degree of the control an agent *has* over the *underlying environment state*. Our approach seeks to infer and gain control over a representation of the environment's state, as opposed to the agent's observations. We demonstrate environments where minimizing observational surprise and maximizing empowerment leads to degenerate solutions that ignore important factors of variation, whereas our approach identifies and controls them. Representation learning methods have been explored in a variety of prior work, including, but not limited to, [39, 72, 29, 13, 51, 76, 22, 40]. Our approach employs a representation learning method to build a latent state-space model [72, 38, 29, 22, 47, 73, 71, 40, 10, 23, 48, 58].

## 6   Discussion

We presented IC2, a method for intrinsically motivating an agent to discover, represent, and exercise control of dynamic objects in a partially-observed environments sensed with visual observations. We found that our method approached expert-level performance on several environments and substantially surpassed prior work in its unsupervised control capability. While our experiments represent a proof-of-concept that illustrates how latent state belief entropy minimization can incentivize an agent to both gather information and gain control over its environment, there are a number of exciting future directions. First, our method is inspired by a connection between thermodynamics and information theory, but the treatment of this connection is informal. Formalizing this connection could lead to an improved theoretical understanding of how IC2 and other intrinsic motivation methods can lead to desirable behavior, and perhaps allow deriving conditions on environments under which such desirable behaviors would emerge; such a characterization might invoke the fact that the Certainty reward degenerates to activate state estimation unless the environment contain sources of uncertainty outside of the robot's state. Second, a precise characterization of environments in which it may not be desirable to deploy IC2 would be useful, for example, (i) if the Niche-based approaches were deployed simultaneously on multiple robots in the same environment, each robot's goal would likely learn to trap or disable the other robots, in order to collapse its uncertainty about them, which may be undesirable, or (ii) if the Niche-based approaches were deployed without safeguards in an environment with living things, similarly, the agent may learn to trap or destroy them, unless the agent estimated it could better avoid future uncertainty by cooperating with them. Finally, since IC2 relies on the LSSM to estimate beliefs, it is limited by the capacity of LSSMs to properly model beliefs – an exciting future direction and an active area of research is extending the capabilities of LSSMs, which would enable study of the behavior that emerges when IC2 is deployed in more complex environments.

**Acknowledgements**  This research was supported by ARL DCIST CRA W911NF-17-2-0181, the Office of Naval Research, and an NSERC Vanier Canada Graduate Scholarship.

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
