# OpenReview forum: "Information is Power: Intrinsic Control via Information Capture"
_NeurIPS.cc/2021/Conference — NeurIPS 2021 Poster_

### Official Review · Reviewer_48zc · 2021-07-15

**Rating:** 7
**Confidence:** 3

**Summary:**

This paper studies the problem of designing intrinsic rewards that encourages active exploration in partially observable dynamic environments. The authors observe that by minimizing the entropy of the latent visitation frequency, the agent is able to achieve both the active information gathering and prevent the environment from changing. Specifically, the authors introduce Believer, a novel framework based on variational latent Bayes filters with discrete latent states, and designed several rewards. In the experiments, Believer has achieved good performance on 3 different tasks.

**Limitations And Societal Impact:**

1/ I think the current experiments are all conducted in relatively simple domains. It would be good to see Believer applied to more complex domains, e.g., robot navigation in REALISTIC environments. The current experiments have small spaces, and it would be good to see if the proposed intrinsic reward is still good for much more complex environments. In addition, a minor point is that it is a bit unclear if the proposed LSSM is still sufficient for modeling the complex environments.

2/ Is it possible to visualize the learned latent states under different intrinsic rewards? I think it would be nice to have some visualizations of how the beliefs are being updated through time.

3/ How does the size of the latent belief affect the performance? Will a larger belief size keep improving the performance? Or is it possible that with a slightly bigger belief size, other variants also achieve comparable performance with Believer?

4/ Some notations are undefined. For example, in line 107 and 108, \hat{p}(o) and \hat{p}(s) are undefined.


**Main Review:**

I think this is an interesting paper with good insights. Reinforcement learning in partially observable environments is always a challenging task, and a good intrinsic reward can significantly improve the exploration and the overall performance of the system. Believer has made a good observation of using the latent visitation frequency for encouraging both the active information gathering and interacting with the environment. Furthermore, the discrete latent states make evaluating the visitation frequency possible.

Overall, I think the paper has made enough contributions. I have some minor suggestions about the presentation and the experiments. Please find detailed comments below.


**Time Spent Reviewing:**

6

---

> ### Author Response · Authors · 2021-08-10
> **Author response to reviewer 48zc**
>
> > I think the current experiments are all conducted in relatively simple domains. It would be good to see Believer applied to more complex domains, e.g., robot navigation in REALISTIC environments. The current experiments have small spaces, and it would be good to see if the proposed intrinsic reward is still good for much more complex environments. In addition, a minor point is that it is a bit unclear if the proposed LSSM is still sufficient for modeling the complex environments.
>
> We agree that scaling up our method to more realistic environments is a very interesting direction for future work! Indeed, it is likely that this would present challenges to the LSSM, and we agree with the reviewer that this is the most likely point of failure when scaling up. However, the focus of our work is on demonstrating the core concept and realization of intrinsic control in CHMPs via our method. For this reason, we would leave scaling up LSSMs and our method to realistic settings for future work, which likely would be feasible as LSSMs improve more generally. We will add discussion of this as future work and note limitations of current LSSMs in the discussion section.
>
> > Is it possible to visualize the learned latent states under different intrinsic rewards? I think it would be nice to have some visualizations of how the beliefs are being updated through time.
>
> Yes, we will include more visualizations of the model’s visual predictions in the supplementary material for each of the intrinsic reward signals.
>
> > How does the size of the latent belief affect the performance? Will a larger belief size keep improving the performance? Or is it possible that with a slightly bigger belief size, other variants also achieve comparable performance with Believer?
>
> Thank you for this question. Early in our experimentation, we experimented briefly on the TwoRoom environment with different sizes of the latent state, and found that varying it by halving and doubling it did not substantially affect performance. We will include the full quantitative results in the experiments section if space permits; otherwise we will include them in the appendix and reference them in the experiments section.
>
> >  Some notations are undefined. For example, in line 107 and 108, \hat{p}(o) and \hat{p}(s) are undefined.
>
> Thank you for noticing this, we will clarify that the "$\hat{\phantom{\cdot}}$" character denotes a learned model.

---

### Official Review · Reviewer_xeVZ · 2021-07-16

**Rating:** 7
**Confidence:** 4

**Summary:**

This paper presents a set of novel objectives for intrinsic motivation that aim to encourage an agent to learn to exercise control over its environment, despite only having partial observability. The key motivating idea is that the agent be encouraged to reduce "Belief Entropy" by limiting the ability of the environment to change, particularly when that part of the environment is not being observed. Four different objectives are presented that encourage the agent to learn policies that achieve this goal and results are demonstrated on a handful of toy environments meant to illustrate the central claims of the paper. Among the results is good performance on the DefendTheCenter VisDoom task, in which the agent learns to shoot enemies without specific instruction to do so, as this prevents the enemies from continuing to exist and "surprising" the agent in the future.

**Limitations And Societal Impact:**

Though the limitations and societal impact are touched upon in the supplementary materials, it is worth expanding on the impacts in a bit more detail. In particular, it would be meaningful to briefly expand on some example environments in which the proposed method would be likely to result in undesired or dangerous behavior.

Edit: The authors have agreed to address this point in their updated draft, and I am satisfy by their reply in this regard.

**Main Review:**

*High-level Comments*

Overall the paper is clearly written and motivates an interesting problem and proposes a handful of intrinsic objectives that are shown to encourage the desired behavior. Yet while the core narrative of the paper is mostly solid (excepting some of the more minor comments below), the results are lacking and somewhat unconvincing. In particular, the results section is too short, and the results themselves do not support the authors' narrative in a way that leads to clear recommendations about when their approach (or which of their approaches) will perform better than others and when. The discussion of the results (aside from the tables, figures and captions themselves) exists across only a single paragraph in , problematically short considering that a number of the results do not "speak for themselves". There are a few such instances:

- Across all the different environments shown, the Niche Creation + Infogain and the Niche Expansion objectives compete for the best performance, yet /why/ each objectives performs better or worse or how their behaviors differ in a way that could lead to a recommendation across environment types is not explored. In particular, the number of monster kills for the DefendTheCenter environment is unintuitively low compared to the Random Policy (and *zero* for the NC+I objective) despite that killing monsters is described as behavior that we might expect across all policies. This behavior should be discussed.
- Second, the performance on the TwoRoom-Large environment includes only the Niche Creation + Infogain results. This is strange, seeing as it is the only environment for which this is that case.
- Relatedly, the text highlights that the Niche Expansion performs well for the DefendTheCenter but it is unclear why this is the case or what the reader should hope to understand, important details to include if the goal is to prescribe how this approach can be used across environments.

Expanding on the results section to address these points and to better understand when and why each objective is well-suited to its task would greatly strengthen the paper.

*Smaller comments and suggestions*

- To the best of my knowledge, the nature of the "oracle policy" is not elaborated upon. It is unclear how an agent with "access to privileged about the environment state" would achieve good behavior on the objectives of interest, which seek to maintain control over the belief. This would help to better clarify the results and the relation of the proposed techniques against this oracle.
- The addition of the exploratory data and policy intuitively seems an important part of the ability of the chosen approach to succeed. Yet it appears that the approaches the authors compare against (SMiRL and Empowerment) do not include such a mechanism to augment data collection. It may help understanding if the authors comment on how this may influence the comparison between the various approaches.
- The limitations of the approach should not exist only in the supplementary material. In particular, some discussion should be included in the paper regarding in what environments the proposed approach would be expected to either perform poorly or result in undesirable behavior would be a welcome addition.

Edit: The authors have provided significant additional detail in response to my questions and I am satisfied by the depth of their investigation. While I still feel as if the approach has some limitations (and the authors have agreed to expand the discussion of these in the relevant section of their paper) I do not think these limitations diminish the quality of the work. I was also persuaded by the other reviews that I was perhaps too harsh on these grounds. I have updated my score accordingly.

**Time Spent Reviewing:**

7

---

> ### Author Response · Authors · 2021-08-10
> **Author response to reviewer xeVZ**
>
> Thank you for the review and constructive comments! Below, we address each of your concerns: we added analysis of experimental results in the DefendTheCenter environment to better study the reasons for the different performance of various methods, added the remaining methods in the TwoRoomLarge environment, added a recommendation of the best-performing method, added a description of the oracle policies, clarified that all policies follow the same exploration procedure, and added additional discussion of limitations and impact. We believe that, with these additions, we have addressed all of the concerns raised in your review. Please let us know if any issues remain!
>
> > The results section is too short…the number of monster kills for the DefendTheCenter environment is unintuitively low compared to the Random Policy (and zero for the NC+I objective) despite that killing monsters is described as behavior that we might expect across all policies…Relatedly, the text highlights that the Niche Expansion performs well for the DefendTheCenter but it is unclear why this is the case
>
> Thank you for your comment, we will expand the discussion of these results by adding written analysis of each of the methods in order for the reader to understand the results in more detail. We have added videos of the observations below to the supplementary website at https://sites.google.com/view/believer-anonymous/home.
> - We observe that the Infogain-based agent pans the camera around until a monster is close and then quickly moves the camera back and forth, centered on the monster, but without shooting at all. Our interpretation of this behavior is that the Infogain reward is causing the agent to create and maintain a large visual stimulus that is difficult for the model to predict due to the stochastic motion of the monster. This interpretation is supported by Table 2, where Infogain and Niche creation + Infogain agents have the highest number of visible monsters in their field of view.
> - We also observe that the Niche Creation + Infogain-based policy exhibits similar behavior to Infogain, which we speculate may be due to the infogain component of the objective dominating the niche creation part.
> - We observe the Niche Creation agent to minimize visibility of the monsters in the following way: the agent tends to shoot the immediately visible monsters upon first spawning and then toggle between turning left and right. Since there is no no-op action, this allows the agent to see as few monsters as possible, since its view is  centered on a region of the map it just cleared.
> - We observe the Niche Expansion agent to typically rotate in one direction and reliably shoot monsters. Our interpretation of this is that the Niche Expansion reward is causing the agent to seek out and reduce the number of surprising monsters more often than the Niche Creation agent due to the belief entropy bonus term that Niche Expansion contains, which encourages more exploration compared to Niche Creation.
> - The Certainty agent has similar behavior to the Niche Creation agent but we observe that it is not as successful and has fewer monster kills and more visible monsters than Niche Creation.
>
> > The performance on the TwoRoom-Large environment includes only the Niche Creation + Infogain results
>
> Thank you for your comment, we will include the complete results for this environment.
>
> | Method | Obj. Lock | Obj. Visible | $H(d^\pi(s_d))$ |
> | ----------- | ----------- |  ----------- |  ----------- |
> |Niche Creation+Infogain| 0.665±0.013 | 0.501±0.013 | 2.831±0.083 |
> |Niche Expansion| 0.002±0.001 | 0.091±0.008 | 3.333±0.006 |
> |Niche Creation | 0.000±0.000 | 0.004±0.001 | 3.322±0.000 |
> |Infogain | 0.262±0.019 | 0.226±0.010 | 3.576±0.043 |
> |Certainty | 0.000±0.000 | 0.004±0.001 | 3.322±0.000 |
> |SMiRL | 0.110±0.020 | 0.087±0.016 | 3.417±0.022 |
> |Random Policy | 0.271±0.024 | 0.138±0.011 | 3.404±0.063 |
> | Oracle | 0.885±0.005 | 0.464±0.017 | 1.016±0.043|
>
> > Clear recommendations about when their approach (or which of their approaches) will perform better than others and when
>
> We will include the following recommendation in the paper. The results show that employing the Niche Creation objective accompanied by a within-episode bonus term -- resulting in either the Niche Expansion objective or the Niche Creation + Infogain objective -- achieves the best control performance across the environments tested.
>
> > To the best of my knowledge, the nature of the "oracle policy" is not elaborated upon. It is unclear how an agent with "access to privileged about the environment state" would achieve good behavior on the objectives of interest, which seek to maintain control over the belief.
>
> Thank you for your comment, we will include this missing information in the updated version. The oracle policies are designed to maximize the control performance in each environment as much as practically possible to put the control performance of our intrinsic objectives in context. Their purpose is to approximate the highest possible performance in each environment, for which access to privileged information can be convenient. In the TwoRoom and OneRoomCapture3D environments, the oracle policies are scripted, using access to the environment state, to seek out the nearest unfrozen object and freeze it, thereby achieving high performance on the lock, capture, and entropy control metrics. In the DefendTheCenter environment, the oracle policy is a neural network trained using PPO to shoot as many monsters as possible by accessing the true extrinsic task reward of the environment.
>
> > The addition of the exploratory data and policy intuitively seems an important part of the ability of the chosen approach to succeed. Yet it appears that the approaches the authors compare against (SMiRL and Empowerment) do not include such a mechanism to augment data collection. It may help understanding if the authors comment on how this may influence the comparison between the various approaches
>
> Thank you for your comment, we will clarify this important point in the paper. The exploratory policy is used to collect additional data for the LSSM only -- the additional data is not added to the training data of the control policy (which we train on-policy with PPO). No method, including ours, is provided with an explicit separate mechanism to augment data collection for control policy optimization. All control policies employed in our methods and the baselines follow the same exploration procedure that occurs as part of PPO optimization.
>
> > The limitations of the approach should not exist only in the supplementary material. In particular, some discussion should be included in the paper regarding in what environments the proposed approach would be expected to either perform poorly or result in undesirable behavior would be a welcome addition
>
> Thank you for your comment, we will make sure that an expanded discussion of limitations is included in the main body of the paper. There are two types of settings in which our environment may exhibit undesirable behavior: 1. settings in which the controlling behavior is desirable and unobtainable; 2. settings in which the controlling behavior is undesirable and obtainable.
> 1) One limitation of our method is that because it relies on the LSSM to estimate beliefs, it is limited by the capacity of LSSMs to properly model beliefs. While a separate analysis of this capacity is outside the scope of the paper, we expect that very complex environments, such as large real-world environments, may be beyond the capabilities of current LSSMs. Of course, as better LSSMs are developed (which is an active area of research), this limitation may be mitigated.
> 2) In addition to the Niche-based methods simplifying to SMiRL-like behavior in fully-observed environments as per the existing discussion, in many partially-observed environments, the only uncertainty is in the robot’s state itself. As discussed on L137-L141, the Certainty reward should result in active state estimation, and other rewards that reduce the entropy of the latent visitation would likely result in a mix of active state estimation while minimizing the amount of motion of the robot. This behavior would perhaps only be undesirable because it may not be very interesting. Thus, the approach is of particular interest in environments that contain sources of uncertainty outside of the robot’s state itself, in which it should gather information to reduce these uncertainties.
>
> > It would be meaningful to briefly expand on some example environments in which the proposed method would be likely to result in undesired or dangerous behavior.
>
> A few examples in which it would be undesirable for the robot to reduce its uncertainty in entities outside itself are: (i) If the Niche-based approaches deployed simultaneously on multiple robots in the same environment, each robot’s goal would likely learn to trap or disable the other robots, in order to collapse its uncertainty about them, which may be undesirable. (ii) If the Niche-based approaches were deployed without safeguards in an environment with living things, similarly, it may learn to trap or destroy them, unless the agent estimated it could better avoid future uncertainty by cooperating with them. We will include these in the paper.

---

> > ### Author Response · Authors · 2021-08-16
> > **Recap of author response to reviewer xeVZ**
> >
> > Hello reviewer xeVZ, we would be grateful if you can confirm whether our response addressed your concerns, and let us know if any issues remain. To recap our response, we:
> > - Added analysis of experimental results in the DefendTheCenter environment to better study the reasons for the different performance of various methods
> > - Added results on the remaining methods in the TwoRoomLarge environment
> > - Added a recommendation of the best-performing method
> > - Added a description of the oracle policies
> > - Clarified that all policies follow the same exploration procedure
> > - Added additional discussion of limitations and impact.

---

> > > ### Comment · Reviewer_xeVZ · 2021-08-25
> > > **Thank you for your detailed reply**
> > >
> > > Yes, I feel I understand the methods and results better and I am happy for the additional details provided here. I will need additional time to fully digest the new details, but they seem to address many (if not all) of my comments and concerns. I will be sure to re-evaluate my review in light of these additions when discussing with the other reviewers.

---

> > > > ### Author Response · Authors · 2021-08-30
> > > > **Thank you for responding; any remaining issues?**
> > > >
> > > > Hello reviewer xeVZ, thank you for taking the time to read and acknowledge our response. We are pleased to learn that our response seems to address many (if not all) of your comments and concerns. Because the discussion period ends this week, we would appreciate it if you could let us know whether our responses are sufficient and whether you have any remaining concerns.

---

### Official Review · Reviewer_6Sqq · 2021-07-16

**Rating:** 7
**Confidence:** 4

**Summary:**

This paper introduces a new intrinsic reward signal for partially observable environments with dynamic objects that can potentially be controlled by an agent. This reward seeks to minimize the entropy of the agent's state visitation as estimated by a latent-space model.
To do so it makes use of a latent state estimation approach using a VAE and an ensemble of dynamics models, and specifies 4 reward functions that can be seen as optimizing the above objective through different means.

These proposed reward functions are evaluated on three domains with partial observability and dynamic objects that the agent's actions can somehow affect and potentially control. The experiments show that one or the other of the proposed reward functions lead to the agent controlling these dynamic objects, whereas previously proposed rewards fail to do so. Of note in the VizDoom domain, the paper shows that one of the proposed rewards leads to maximization of the extrinsic reward without learning to do so.

**Limitations And Societal Impact:**

Discussed in supplementary material.

**Main Review:**

## Technical Positives:
* Taking the problem formulation used for intrinsic control to the harder domain of partial observability with dynamic objects is a step forward towards more complex domains.
* This paper identifies what intrinsic control would mean in such an environment and contrasts it with approaches that have been used in the fully observable setting.
* It then proposes reward functions that would lead to such control being exerted by the agent, by manipulating the entropy of its state belief and visitation.
* These rewards seem to be validated in the three domains the paper evaluates on, though in practice it seems like the Niche expansion and Niche Creation + Infogain objectives are the ones that show results that are different and improvements on the baselines.

## Questions, Remarks, and Drawbacks:
* Since the proposed objective seeks to reduce the entropy of its state visitation, if left to its own devices it seems like the approach would also learn to go to a part of the environment where it does not encounter any dynamic objects and not move. The control aspect seems to arise from a well trained latent state model whose components include the encoding from observations to the latent state (which is explained appropriately in the paper) but also an exploration policy that learns both an accurate estimate of the latent state as well as an accurate estimate of the uncertainty of the dynamic objects. This exploration policy seems to not have been emphasized and explained enough in the paper.
* Is the same exploration policy used to collect data for the baseline methods? Would those methods benefit from a similar exploration?
* SMiRL is proposed as an intrinsic reward in CMPs that proposes rewards on the states encountered. This paper argues that the equivalent in the CHMP would be rewards computed based on observations. However I would argue that a reasonable equivalent would be to compute the SMiRL reward on the state estimated by the latent state encoder. Why is considering the observations a more appropriate analog for SMiRL? If the state used for estimating the reward in SMiRL is the latent state, then do the differences between the proposed rewards and SMiRL become more nebulous?
* In Figure 4 (b), it would be helpful if the reward functions depicted are further labeled with the names in Section 5.1, so that correspondence can be drawn more readily.

**Time Spent Reviewing:**

8

---

> ### Author Response · Authors · 2021-08-10
> **Author response to reviewer 6Sqq**
>
> Thank you for the review and constructive comments! Below, we address each of your concerns: we include additional explanation of the exploration policy, we clarify how SMiRL relates to the proposed approach, we clarify the role of exploration in our method and the baselines, and we incorporate your advice on improving Fig 4b.
>
> > Since the proposed objective seeks to reduce the entropy of its state visitation, if left to its own devices it seems like the approach would also learn to go to a part of the environment where it does not encounter any dynamic objects and not move. The control aspect seems to arise from a well trained latent state model whose components include the encoding from observations to the latent state (which is explained appropriately in the paper) but also an exploration policy that learns both an accurate estimate of the latent state as well as an accurate estimate of the uncertainty of the dynamic objects. This exploration policy seems to not have been emphasized and explained enough in the paper.
>
> Thank you for raising this point. If the agent is unable to encounter or represent dynamic objects, it will choose not to move. However, if the model properly captures the existence and motion of dynamic objects (even out of view), then the control policy has an incentive to exert control over them, rather than avoid them by going to parts of the environment where it does not encounter them. Thus, the optimal policy equipped with the optimal model will seek out and control all dynamic objects. The exploration policy is employed as a way to encounter dynamic objects and collect data on which the model performs poorly. The exploration policy does not need to learn accurate estimates of the latent state and its uncertainty -- it is merely incentivized to visit where the current model is inaccurate. We will expand the existing discussion of this point in L151-153 to include these points.
>
> > SMiRL is proposed as an intrinsic reward in CMPs that proposes rewards on the states encountered. This paper argues that the equivalent in the CHMP would be rewards computed based on observations. However I would argue that a reasonable equivalent would be to compute the SMiRL reward on the state estimated by the latent state encoder. Why is considering the observations a more appropriate analog for SMiRL? If the state used for estimating the reward in SMiRL is the latent state, then do the differences between the proposed rewards and SMiRL become more nebulous?
>
> Thank you for raising this point. Indeed, it is a matter of interpretation of what the most natural extension of SMiRL to the partially-observed setting is -- whether (1) the observation should be treated as the state, or (2) an LSSM’s belief should be treated as an estimate of the latent state. Thus, our experiments include both interpretations, the latter constituting one of our methods. (1) Although SMiRL assumes full observability and was primarily applied to the fully-observed setting, it was also applied to a partially-observed setting by treating the observation as the state -- see the HauntedHouse environment depicted in Fig. 5 of SMiRL, which shows SMiRL learning to avoid dynamic objects, similar to how it did so in our experiments. Thus, one interpretation of SMiRL is to operate directly on the observations. One of the purposes of including this interpretation in the experiments is to show how observational entropy-based SMiRL in CHMPs fails to exhibit control. (2) A second interpretation looks similar to our method, as the reviewer suggested: the Niche Creation objective may be considered to be “SMiRL applied to samples of latent beliefs estimated from a LSSM in a CHMP.” We will incorporate parts of this discussion into L111-113 in order to discuss both interpretations.
>
> > Is the same exploration policy used to collect data for the baseline methods? Would those methods benefit from a similar exploration?
>
> We will make sure to clarify this point in the paper. The exploration policy is a component of our method that depends on the LSSM and is used only to improve the LSSM -- the data collected by the exploration policy is not in the on-policy learning of the control policy (although it could be used, in principle, by an off-policy learning procedure). No method, including ours, is provided with an explicit separate mechanism to augment data collection for policy optimization. All control policies employed in our methods and the baselines follow the same exploration procedure that occurs as part of PPO optimization.
>
> > In Figure 4 (b), it would be helpful if the reward functions depicted are further labeled with the names in Section 5.1, so that correspondence can be drawn more readily.
>
> Thank you for your comment, we agree, and will do so.

---

> > ### Comment · Reviewer_6Sqq · 2021-08-31
> > **Re: Response**
> >
> > Thank you to the authors for clarifying the points above.
> >
> > If these clarifications are added into the paper I believe the juxtaposition with SMiRL will be clearer, and the contributions of this work will be more impactful. I am updating my score accordingly.

---

### Official Review · Reviewer_UHCB · 2021-07-25

**Rating:** 8
**Confidence:** 4

**Summary:**

This paper introduces a novel latent state-space method for unsupervised reinforcement learning in dynamic partially-observed visual environments. The key insight of the paper is that minimizing the entropy of the belief over the latent state does not incentivize agents to construct a "niche", i.e. modify the environment such that it becomes more easily controlled. As a remedy, the authors propose to instead minimise the entropy of the belief over latent state visitation. The authors discuss a variety of reward incentivised constructed from the latent-space model and introduce a separate exploration policy. The authors evaluate on three separate single-agent environments, including one with continuous action spaces. It is found that the novel latent-visitation based reward incentives outperform existing methods on a variety of metrics.

**Limitations And Societal Impact:**

In Table 1 (OneRoomCapture3D), how is it possible that Niche Expansion score significantly lower in entropy, while Niche Creation+Infogain wins wrt both obj capture and obj visible metrics?

The analogy with Maxwell's demon may deserve further detailed explanation: It is not clear to me what exactly the "potential energy"-analogy constitutes.

**Main Review:**

The paper's fundamental idea about minimizing the entropy of the belief order latent-state visitation is original, and clearly marks a step change over previous approaches. The ideas are subsequently studied in appropriate detail and evaluated on a broad set of toy environments. The quality of both theoretical and empirical developments is high.
The paper is written very clearly.

Overall, I believe this paper is very solid and deserves acceptance.






**Time Spent Reviewing:**

3

---

> ### Author Response · Authors · 2021-08-10
> **Author response to reviewer UHCB**
>
> Thank you for the review and constructive comments!
>
> > In Table 1 (OneRoomCapture3D), how is it possible that Niche Expansion score significantly lower in entropy, while Niche Creation+Infogain wins wrt both obj capture and obj visible metrics?
>
> Thank you for this question. Niche Expansion (NE) scores lower (better) than Niche Creation+Infogain (NC+I) in the box entropy metric yet slightly worse in the Captured metric due to (1) differences in how each agent tends to capture the box and (2) higher variance in the NC+I policy optimization. (1) The NE agent tends to moves towards the box and execute the ‘freeze’ action to completely stop the box from moving, whereas the NC+I agent tends to move toward the box and trap it next to the wall without executing the freeze action, which causes the box to bounce back and forth between the wall and the agent every timestep, which is measured as Captured. We hypothesize NC+I does this because allowing the box to retain some movement gives the agent more Infogain reward. (2) The higher variance in the policy optimization of the NC+I agent resulted in some runs that performed worse, which impacted both the entropy and capture metrics, but had a stronger effect on the box capturing metric. We will include discussion of these points in the paper, along with qualitative illustrations of these behaviors.
>
> > The analogy with Maxwell's demon may deserve further detailed explanation: It is not clear to me what exactly the "potential energy"-analogy constitutes.
>
> Thank you for your comment. We will expand the details of discussion to form a clearer analogy in the paper. One detail is that having an ordered, stabilized environment is a useful precondition for certain types of tasks; by ordering the environment first, the effort the agent must expend in the future to accomplish these tasks can be reduced.

---

> > ### Comment · Reviewer_UHCB · 2021-08-29
> > **Reviewer response**
> >
> > Many thanks for your clarifications. My score remains unchanged.

---

### Decision · Program_Chairs · 2021-09-27

**Decision:**

Accept (Poster)

**Comment:**

Learning agents need to be able to explore and control their environment. The authors introduce a novel objective, the entropy of the state visitation, which combines these two desiderata and is tightly connected to information gain for controlling environments.

The authors show promising results on a number of benchmarks created to highlight partial observability and clearly ablate the different components of their algorithm on these environments.

The discussion focussed on a few key points: The authors provided a more detailed comparison to SMiRL, added additional experimental data as requested by the reviewers and clarified technical points. Overall, the reviewers felt that their concerns had been addressed appropriately by the authors.

Given the positive reviews, the discussion and the overall contributions of the paper, I recommend this paper to be accepted.